# Query-efficient Meta Attack to Deep Neural Networks

**Jiawei Du**[1,3]*, **Hu Zhang**[2]*, **Joey Tianyi Zhou**[3], **Yi Yang**[2] , **Jiashi Feng**[1]
[1]Dept. ECE, National University of Singapore, Singapore
[2]ReLER, University of Technology Sydney, Australia
[3]Institute of High performance Computing, A*STAR, Singapore
dujiawei@u.nus.edu,Hu.Zhang-1@student.uts.edu.au
joey.tianyi.zhou@gmail.com,Yi.Yang@uts.edu.au
elefjia@nus.edu.sg

## Abstract

Black-box attack methods aim to infer suitable attack patterns to targeted DNN models by only using output feedback of the models and the corresponding input queries. However, due to lack of prior and inefficiency in leveraging the query and feedback information, existing methods are mostly query-intensive for obtaining effective attack patterns. In this work, we propose a meta attack approach that is capable of attacking a targeted model with much fewer queries. Its high query-efficiency stems from effective utilization of meta learning approaches in learning generalizable prior abstraction from the previously observed attack patterns and exploiting such prior to help infer attack patterns from only a few queries and outputs. Extensive experiments on MNIST, CIFAR10 and tiny-Imagenet demonstrate that our meta-attack method can remarkably reduce the number of model queries without sacrificing the attack performance. Besides, the obtained meta attacker is not restricted to a particular model but can be used easily with a fast adaptive ability to attack a variety of models. The code of our work is available at https://github.com/dydjw9/MetaAttack_ICLR2020/.

## 1 Introduction

Despite the great success in various tasks, deep neural networks (DNNs) are found to be susceptible to adversarial attacks and often suffer dramatic performance degradation in front of adversarial examples, even if only tiny and invisible noise is imposed on the input (Szegedy et al., 2014). To investigate the safety and robustness of DNNs, many adversarial attack methods have been developed, which apply to either a white-box (Goodfellow et al., 2015; Moosavi-Dezfooli et al., 2016; Carlini & Wagner, 2017; Madry et al., 2018) or a black-box setting (Papernot et al., 2017; Brendel et al., 2018; Narodytska & Kasiviswanathan, 2017). In the white-box attack setting, the target model is transparent to the attacker and imperceptible adversarial noise can be easily crafted to mislead this model by leveraging its gradient information (Goodfellow et al., 2015). In contrast, in the black-box setting, the structure and parameters of the target DNN model are invisible, and the adversary can only access the input-output pair in each query. With a sufficient number of queries, black-box methods utilize the returned information to attack the target model generally by estimating gradient (Chen et al., 2017; Ilyas et al., 2018a; Narodytska & Kasiviswanathan, 2017; Cheng et al., 2019).

Black-box attack is more feasible in realistic scenarios than white-box attack but it is much more query-intensive. Such a drawback is largely attributed to the fact that returned information for each queried example is sparse and limited. During inferring attack patterns, existing black-box methods simply integrate the information between two sequential iterations brutally and ignore the implicit but profound message, thus not fully exploiting the returned information. Although query-efficient algorithms for generating attack examples are very meaningful in practice (Ilyas et al., 2018a), how to enhance query-efficiency for black-box attack remains underexplored.

---

*Equal contribution.

In this work, we address a query-efficiency concerned attack problem. Particularly, we consider only top-$k$ probability scores accessible from the target black-box model. With this practical but challenging scenario, we aim at three important objectives: lower query number, higher success rate and smaller noise magnitude. We develop a meta-learning based attack method, which applies meta learning to obtaining prior information from the successful attack patterns, and uses the prior for efficient optimization. Specifically, we propose to train a meta attacker model through meta learning (Nichol et al., 2018), inspired by its success in solving few-shot learning problems. We first deploy several existing classification models to get pairs of (*images, gradients*) with the max-margin logit classification loss. Then we use the data pairs of each classification model to train the meta attacker. After obtaining the attacker, we use it to attack a new black-box model for accelerating the search process for adversarial examples by optimizing it with coordinate-wise gradient estimation. Different from previous methods, we use the estimated gradient not only to update adversarial noise but to fine-tune the well-trained attacker. After few-shot fine-tuning, the attacker is able to simulate the gradient distribution of the target model.

We evaluate our method on MNIST, CIFAR10 and tiny-ImageNet datasets by comparing it with state-of-the-art black-box attack methods including Zoo (Chen et al., 2017), Decision-Boundary (Brendel et al., 2018), AutoZoom (Tu et al., 2019), Opt-attack (Cheng et al., 2019) and Bandits (Ilyas et al., 2018b). In both targeted and untargeted settings, our proposed method achieves comparable attack success rate and adversarial perturbation to all baselines but with a significantly reduced query number. The detailed experiment results demonstrate our superior query-efficiency.

## 2 Related Work

Classical white-box attack methods include Fast-Gradient Sign Method (FGSM) (Goodfellow et al., 2015), IFGSM (Madry et al., 2018), DeepFool (Moosavi-Dezfooli et al., 2016) and C&W attack (Carlini & Wagner, 2017), following a setting where detailed information about the target model (gradients and losses) is provided. Comparatively, the black-box setting better accords with the real world scenarios in that little information about the target model is visible to the attacker. The pioneer work on black-box attack (Papernot et al., 2017) tries to construct a substitute model with augmented data and transfer the black-box attack problem to a white-box one. However, its attack performance is very poor due to the limited transferability of adversarial examples between two different models. (Brendel et al., 2018) considers a more restricted case where only top-1 prediction classes are returned and proposes a random-walk based attack method around the decision boundary. It dispenses class prediction scores and hence requires extensive model queries. Zoo (Chen et al., 2017) is a black-box version of C&W attack, achieving a similar attack success rate and comparable visual quality as many white-box attack methods. However, its coordinate-wise gradient estimation requires extensive model evaluations. More recently, (Ilyas et al., 2018a) proposes a query-limited setting with $L_\infty$ noise considered, and uses a natural evolution strategy (NES) to enhance query efficiency. Though this method successfully controls the query number, the noise imposed is larger than average. (Narodytska & Kasiviswanathan, 2017) proposes a novel local-search based technique to construct numerical approximation to the network gradient, which is then carefully used to construct a small set of pixels in an image to perturb. It suffers a similar problem as in (Chen et al., 2017) for pixel-wise attack. (Cheng et al., 2019) considers a hard-label black-box setting and formulates the problem as real-valued optimization that is solved by a zeroth order optimization algorithm. Ilyas et al. (2018b) reduce the queries by introducing two gradient priors, the time-independent prior and the data-dependent prior, and reformulating the optimization problem.

We then briefly introduce some works on meta-learning related to our work. Meta-learning is a process of learning how to learn. A meta-learning algorithm takes in a distribution of tasks, each being a learning problem, and produces a quick learner that can generalize from a small number of examples. Meta-learning is very popular recently for its fast adaptive ability. MAML (Finn et al., 2017) is the first to propose this idea. Recently, a simplified algorithm Reptile (Nichol et al., 2018) which is an approximation to the first-order MAML is proposed, achieving higher efficiency in computation and consuming less memory. With these superior properties, meta learning is applied to adversarial attack methods (Zgner & Gnnemann, 2019; Edmunds et al., 2017). Zgner & Gnnemann (2019) try to attack the structure of a graph model in the training process to decrease the model generalization performance. Edmunds et al. (2017) investigate the susceptibility of MAML to adversarial attacks and the transferability of the obtained meta model to a specific task.

## 3 METHOD

### 3.1 PRELIMINARIES: BLACK-BOX ATTACK SCHEMES

We first formulate the black-box attack problem and introduce the widely used solutions. We use $(\boldsymbol{x}, t)$ to denote the pair of a natural image and its true label, and $\hat{\boldsymbol{x}}$ and $\hat{t}$ to denote the adversarial perturbed version of $\boldsymbol{x}$ and the returned label by the target classification model $\mathcal{M}_{tar}$. The black-box attack aims to find an adversarial example $\hat{\boldsymbol{x}}$ with imperceivable difference from $\boldsymbol{x}$ to fail the target model , i.e., $\hat{t} \neq t$ through querying the target model for multiple times. It can be formulated as

$$
\begin{aligned}
\min_{\hat{\boldsymbol{x}}} & \ \ell(\hat{\boldsymbol{x}}, \mathcal{M}_{tar}(\hat{\boldsymbol{x}}), t) \\
\text{s.t. } & \|\hat{\boldsymbol{x}} - \boldsymbol{x}\|_p \leq \rho, \ \ \#\text{queries} \leq \text{Q}.
\end{aligned}
\tag{1}
$$

Here $\|\cdot\|_p$ denotes the $\ell_p$ norm that measures how much perturbation is imposed. $\mathcal{M}_{tar}(\hat{\boldsymbol{x}})$ is the returned logit or probability by the target model $\mathcal{M}_{tar}$. The loss function $\ell(\hat{\boldsymbol{x}}, \mathcal{M}_{tar}(\hat{\boldsymbol{x}}), t)$ measures the degree of certainty for model $\mathcal{M}_{tar}$ assigning the input $\hat{\boldsymbol{x}}$ into class $t$. One common used adversarial loss is the probability of class $t$: $\ell(\hat{\boldsymbol{x}}, \mathcal{M}_{tar}(\hat{\boldsymbol{x}}), t) = p_{\mathcal{M}_{tar}}(t|\hat{\boldsymbol{x}})$. The first constraint enforces high similarity between the clean image $\boldsymbol{x}$ and the adversarial one $\boldsymbol{x}_{adv}$ and the second imposes a fixed budget $Q$ for the number of queries allowed in the optimization.

In the white-box attack setting, the adversary can access the true gradient $\nabla_{\hat{\boldsymbol{x}}_t} \ell(\hat{\boldsymbol{x}}_t)$ and perform gradient descent $\hat{\boldsymbol{x}}_{t+1} = \hat{\boldsymbol{x}}_t - \nabla_{\hat{\boldsymbol{x}}_t} \ell(\hat{\boldsymbol{x}}_t)$. But in the black-box setting, the gradient information $\nabla_{\hat{\boldsymbol{x}}_t} \ell(\hat{\boldsymbol{x}}_t)$ is not attainable. In this case, the attacker can estimate the gradient using only queried information from model evaluation such as hard label, logits and probability scores. This kind of estimator is the backbone of so-called zeroth-order optimization approaches (Chen et al., 2017; Narodytska & Kasiviswanathan, 2017; Tu et al., 2019; Ilyas et al., 2018a;b). The estimation is done via finite difference method (Chen et al., 2017; Narodytska & Kasiviswanathan, 2017; Tu et al., 2019), which finds the $k$ components of the gradient by estimating the inner products of the gradients with all the standard basis vector $\boldsymbol{e}_1, ..., \boldsymbol{e}_k$:

$$
\nabla \ell(\boldsymbol{x}) \approx \sum_{i=1}^{k} \frac{f(\boldsymbol{x} + h\boldsymbol{e}_i) - f(\boldsymbol{x} - h\boldsymbol{e}_i)}{2h} \boldsymbol{e}_i,
\tag{2}
$$

where step size $h$ controls the quality of the estimated gradient. Another strategy is to reformulate the loss function (Ilyas et al., 2018a;b). Instead of computing the gradient of $\ell(\boldsymbol{x})$ itself, the expected value of loss function $\ell(\boldsymbol{x})$ under the search distribution is minimized and when the search distribution of random Guassian noise is adopted, the gradient estimation problem transfers into a zeroth-order estimation problem,

$$
\nabla \mathbb{E}[\ell(\boldsymbol{x})] \approx \frac{1}{\sigma n} \sum_{i=1}^{n} \ell(\boldsymbol{x} + \sigma \boldsymbol{\delta}_i) \boldsymbol{\delta}_i
\tag{3}
$$

where $n$ is the amount of noise sampled from the distribution. After obtaining the estimated gradient, classical optimization algorithms (Nesterov, 2013; Johnson & Zhang, 2013) can be used to infer the adversarial examples. Though the estimated gradient may not be accurate, it is still proved useful enough in adversarial attack. The convergence of these zeroth-order methods is guaranteed under mild assumptions (Ghadimi & Lan, 2013; Nesterov & Spokoiny, 2017; Hazan et al., 2016).

Since each model evaluation consumes a query, naively applying the above gradient estimation to black-box attack is quite query expensive due to its coordinate or noise sampling nature. Take the first strategy on tiny-Imagenet dataset for example. It consumes more than 20,000 queries for each image to obtain a full gradient estimate, which is not affordable in practice. In this work, we address such a limitation via developing a query-efficient meta-learning based attack model.

### 3.2 LEARNING OF META ATTACKER

To reduce the query cost for black-box attack, we apply meta learning to training a meta attacker model, inspired by its recent success in few-shot learning problems (Finn et al., 2017; Nichol et al., 2018). The meta attacker learns to extract useful prior information of the gradient of a variety of models w.r.t. specific input samples. It can infer the gradient for a new target model using only a

---

**Algorithm 1** Meta Attacker Training

---

**Input:** Input images $\mathbb{X}$, groundtruth gradients $\mathbb{G}_i$ generated from classification models $\mathcal{M}_i$ to serve as task $\mathcal{T}_i$;
1: Randomly initialize $\boldsymbol{\theta}$;
2: **while** not done **do**
3:      **for** *all* $\mathcal{T}_i$ **do**
4:          Sample $K$ samples from $(\mathbb{X}, \mathbb{G}_i)$ for training, denoted as $(\mathbb{X}_s, \mathbb{G}_i^s)$;
5:          Evaluate $\nabla_{\boldsymbol{\theta}}\mathcal{L}_i(\mathcal{A}_{\boldsymbol{\theta}}) = \nabla_{\boldsymbol{\theta}}\|\mathcal{A}_{\boldsymbol{\theta}}(\mathbb{X}_s) - \mathbb{G}_i^s\|_2^2$ with respect to $(\mathbb{X}_s, \mathbb{G}_i^s)$;
6:          Update $\boldsymbol{\theta}_i' := \boldsymbol{\theta} - \alpha\nabla_{\boldsymbol{\theta}}\mathcal{L}_i(\mathcal{A}_{\boldsymbol{\theta}})$;
7:      **end for**
8:      Update $\boldsymbol{\theta} := \boldsymbol{\theta} + \epsilon\frac{1}{n}\sum_{i=1}^{n}(\boldsymbol{\theta}_i' - \boldsymbol{\theta})$;
9: **end while**
**Output:** Parameters $\boldsymbol{\theta}$ of meta model $\mathcal{A}$.

---

few queries. After obtaining such a meta attacker, we replace the zeroth-order gradient estimation in traditional black box attack methods with it to directly estimate the gradient.

We collect a set of existing classification models $\mathcal{M}_1, ..., \mathcal{M}_n$ to generate gradient information for universal meta attacker training. Specifically, we feed each image $\boldsymbol{x}$ into the models $\mathcal{M}_1, ..., \mathcal{M}_n$ respectively and compute losses $\ell_1, ..., \ell_n$ by using following max-margin logit classification loss:

$$\ell_i(\boldsymbol{x}) = \max\left[\log[\mathcal{M}_i(\boldsymbol{x})]_t - \max_{j\neq t}\log[\mathcal{M}_i(\boldsymbol{x})]_j, 0\right]. \tag{4}$$

Here $t$ is the groundtruth label and $j$ indexes other classes. $[\mathcal{M}_i(\boldsymbol{x})]_t$ is the probability score of the true label predicted by the model $\mathcal{M}_i$, and $[\mathcal{M}_i(\boldsymbol{x})]_j$ denotes the probability scores of other classes. By performing one step back-propagation of losses $\ell_1, ..., \ell_n$ w.r.t. the input images $\boldsymbol{x}$, the corresponding gradients $\boldsymbol{g}_i = \nabla_{\boldsymbol{x}}\ell_i(\boldsymbol{x}), i = 1, ..., n$ are obtained. Finally, we collect $n$ groups of data $\mathbb{X} = \{\boldsymbol{x}\}, \mathbb{G}_i = \{\boldsymbol{g}_i\}, i = 1, ..., n$ to train the universal meta attacker.

We design a meta attacker $\mathcal{A}$ which has a similar structure as an autoencoder, consisting of symmetric convolution and de-convolution layers and outputs a gradient map with the same size as the input. Meta attacker model $\mathcal{A}$ is parameterized with parameters $\boldsymbol{\theta}$.

Due to the intrinsic difference between selected classification models, each obtained set $(\mathbb{X}, \mathbb{G}_i)$ is treated as a task $\mathcal{T}_i$ in meta attacker training. During the training process, for each iteration we only draw $K$ samples from task $\mathcal{T}_i$ and feedback the loss $\mathcal{L}_i$ to update model parameters from $\boldsymbol{\theta}$ to $\boldsymbol{\theta}_i'$. $\boldsymbol{\theta}_i'$ is then computed through one or multiple gradient descents: $\boldsymbol{\theta}_i' := \boldsymbol{\theta} - \alpha\nabla_{\boldsymbol{\theta}}\mathcal{L}_i(\mathcal{A}_{\boldsymbol{\theta}})$. For a sensitive position of the meta attacker, the meta attacker parameters are optimized by combining each $\boldsymbol{\theta}_i'$ across all tasks $\{\mathcal{T}_i\}_{i=1,...,n}$, following the update strategy of Reptile (Nichol et al., 2018) in meta learning,

$$\boldsymbol{\theta} := \boldsymbol{\theta} + \epsilon\frac{1}{n}\sum_{i=1}^{n}(\boldsymbol{\theta}_i' - \boldsymbol{\theta}). \tag{5}$$

We adopt mean-squared error (MSE) as the training loss in the inner update,

$$\mathcal{L}_i(\mathcal{A}_{\boldsymbol{\theta}}) = \|\mathcal{A}_{\boldsymbol{\theta}}(\mathbb{X}_s) - \mathbb{G}_i^s\|_2^2. \tag{6}$$

The set $(\mathbb{X}_s, \mathbb{G}_i^s)$ denotes the $K$ samples used for each inner update from $\boldsymbol{\theta}$ to $\boldsymbol{\theta}_i'$. Since the number of $K$ sampled each time is very small, the update strategy above tries to find good meta attacker parameters $\boldsymbol{\theta}$ as an initial point, from which the meta attacker model can fast adapt to new data distribution through gradient descent based fine-tuning within limited samples. Therefore, this characteristic can be naturally leveraged in attacking new black-box models by estimating their gradient information through a few queries. Detailed training process of our meta attacker is described in Algorithm 1.

### 3.3 QUERY-EFFICIENT ATTACK VIA META ATTACKER

An effective adversarial attack relies on optimizing the loss function equation 1 w.r.t. the input image to find the adversarial example of the target model $\mathcal{M}_{tar}$. Differently, our proposed method applies the meta attacker $\mathcal{A}$ to predicting the gradient map of a test image directly.

---

**Algorithm 2** Adversarial Meta Attack Algorithm

---

**Input:** Test image $x_0$ with label $t$, meta attacker $\mathcal{A}_{\boldsymbol{\theta}}$, target model $\mathcal{M}_{tar}$, iteration interval $m$, selected top-$q$ coordinates;

1: **for** $t = 0, 1, 2, ...$ **do**
2:     **if** $(t+1) \mod m = 0$ **then**
3:         Perform zeroth-order gradient estimation on top $q$ coordinates, denoted as $I_t$ and obtain $g_t$;
4:         Fine-tune meta attacker $\mathcal{A}$ with $(x_t, g_t)$ on $I_t$ by loss $L = \|[\mathcal{A}_{\boldsymbol{\theta}}(x_t)]_{I_t} - [g_t]_{I_t}\|_2^2$;
5:     **else**
6:         Generate the gradient map $g_t$ directly from meta attacker $\mathcal{A}$ with $x_t$, select coordinates $I_t$;
7:     **end if**
8:     Update $[x']_{I_t} = [x_t]_{I_t} + \beta[g_t]_{I_t}$;
9:     **if** $\mathcal{M}_{tar}(x') \neq t$ **then**
10:        $x_{adv} = x'$;
11:        break;
12:     **else**
13:        $x_{t+1} = x'$;
14:     **end if**
15: **end for**

**Output:** adversarial example $x_{adv}$.

---

We use the obtained meta attacker model $\mathcal{A}$ to predict useful gradient map for attacking, which should be fine tuned to adapt to the new gradient distribution under our new target model. Particularly, instead of finetuning once, for each given image $x$, we update $\mathcal{A}$ by leveraging query information with the following periodic scheme. Suppose the given image is perturbed to $x_t \in \mathbb{R}^p$ at iteration $t$. If $(t+1) \mod m = 0$, our method performs zeroth-order gradient estimation to obtain gradient map $g_t$ for fine-tuning. As each pixel value for estimated gradient map consumes two queries, for further saving queries, we just select $q$ of the $p$ coordinates to estimate, $q \ll p$, instead of the full gradient map through all $p$ coordinates. The indexes of chosen coordinates are determined by the gradient map $g_{t-1}$ obtained in iteration $t-1$. We sort the coordinate indexes by the value of $g_{t-1}$ and select top-$q$ indexes. The set of these indexes are denoted as $I_t$. We feed image $x_t$ in iteration $t$ into meta attacker $\mathcal{A}$ and compute the MSE loss on indexes $I_t$, i.e. $L = \|[\mathcal{A}_{\boldsymbol{\theta}}(x_t)]_{I_t} - [g_t]_{I_t}\|_2^2$. Then we perform gradient descent for the MSE loss with a few steps to update the parameters $\boldsymbol{\theta}$ of meta attacker $\mathcal{A}$. For the rest iterations, we just use the periodically updated attacker $\mathcal{A}_{\boldsymbol{\theta}}$ to directly generate the gradient $g_t = \mathcal{A}_{\boldsymbol{\theta}}(x_t)$. When we have the estimated gradient map $g_t$ in iteration $t$, we update to get the adversarial sample $x'_t$ by $[x'_t]_{I_t} = [x_t]_{I_t} + \beta[g_t]_{I_t}$ where $\beta$ is a hyperparameter to be tuned. The details are summarized in Algorithm 2.

In our method, the following operations contribute to reducing the query number needed by the attacker. First, though we just use $q$ coordinates to fine-tune our meta attacker $\mathcal{A}$ every $m$ iterations, the meta attacker $\mathcal{A}$ is trained to ensure that it can abstract the gradient distribution of different $x_t$ and learn to predict the gradient from a few samples with simple fine-tuning. Secondly, the most query-consuming part lies in zeroth-order gradient estimation, due to its coordinate-wise nature. In our algorithm, we only do this every $m$ iterations. When we use the finetuned meta attacker $\mathcal{A}$ directly, no query is consumed in gradient estimation in these iterations. Intuitively, larger $m$ implies less gradient estimation computation and fewer queries. Besides, just as mentioned above, even in zeroth-order gradient estimation, only top-$q$ coordinates are required. Normally, $q$ is much smaller than dimension $p$ of the input.

## 4 EXPERIMENTS

We compare our meta attacker with state-of-the-art black-box attack methods including Zoo (Chen et al., 2017), Decision-Boundary (Brendel et al., 2018), AutoZoom (Tu et al., 2019), Opt-attack (Cheng et al., 2019), FW-black (Chen et al., 2018) and Bandits (Ilyas et al., 2018b) to evaluate its query efficiency. We also study its generalizability and transferability through a *Meta transfer* attacker, as detailed in the following sections.

### 4.1 SETTINGS

**Datasets and Target Models** We evaluate the attack performance on MNIST (LeCun, 1998) for handwritten digit recognition, CIFAR10 (Krizhevsky & Hinton, 2009) and tiny-Imagenet (Russakovsky et al., 2015) for object classification. The architecture details of meta attack models on MNIST, CIFAR10 and tiny-Imagenet are given in Table 6. For MNIST, we train a separate meta attacker model since the images have different channel numbers from other natural image datasets. For CIFAR10 and tiny-Imagenet, we use a common meta attacker model. On CIFAR-10, we choose ResNet18 (He et al., 2016) as the target model $\mathcal{M}_{tar}$ and use VGG13, VGG16 (Simonyan & Zisserman, 2014) and GoogleNet (Szegedy et al., 2015) for training our meta attacker. On tiny-Imagenet, we choose VGG19 and ResNet34 as the target model separately, and use VGG13, VGG16 and ResNet18 for training the meta attacker together.

**Attack Protocols** For a target black-box model $\mathcal{M}_{tar}$ , obtaining a pair of (*input-output*) is considered as one query. We use the mis-classification rate as attack success rate; we randomly select 1000 images from each dataset as test images. To evaluate overall noise added by the attack methods, we use the mean $L_2$ distance across all the samples $noise(\mathcal{M}_{tar}) = \frac{1}{n} \sum_{i=1}^{n} \|\boldsymbol{x}_{i,\mathcal{A},\mathcal{M}_{tar}}^{adv} - \boldsymbol{x}_i\|_2$, where $\boldsymbol{x}_{i,\mathcal{A},\mathcal{M}_{tar}}^{adv}$ denotes the adversarial version for the authentic sample $\boldsymbol{x}_i$.

**Meta-training Details** For all the experiments, we use the same architecture for the meta attacker $\mathcal{A}$ as shown in Table 6. We use Reptile (Nichol et al., 2018) with $0.01$ learning rate to train meta attackers. We use 10000 randomly selected images from the training set to train the meta-attackers in three datasets. The proportion of the selected images to the whole training set are 16%, 20%, and 10% respectively. Fine-tuning parameters are set as $m = 5$ for MNIST and CIFAR10, and $m = 3$ for tiny-Imagenet. Top $q = 128$ coordinates are selected as part coordinates for attacker fine-tuning and model attacking on MNIST; and $q = 500$ on CIFAR10 and tiny-Imagenet.

### 4.2 COMPARISON WITH BASELINES

We compare our meta attacker with baselines for both the untargeted and targeted black-box attack on the three datasets. The results are reported in detail as below.

**Untargeted Attack** Untargeted attack aims to generate adversarial examples that would be misclassified by the attacked model into any category different from the ground truth one. The overall results are shown in Table 1, in which, *Meta transfer* denotes that meta attacker trained on one dataset is used to attack target models on another dataset . Our method is competitive with baselines in terms of adversarial perturbation and success rate, but our query number is reduced.

We also compare the results of our method with Zoo (Chen et al., 2017) and AutoZoom (Tu et al., 2019) from a query-efficiency perspective. We use these models to conduct untargeted attack on CIFAR10 and tiny-Imagenet by limiting a maximum number of queries for each adversarial example and compare their success rate. The results are shown in Fig. 1. We notice that for different query thresholds, the success rate of our method is always higher than Zoo and AutoZoom. This is possibly because the testing samples have different $L_2$ distances to the decision boundary. Higher success rate of our method indicates our meta attacker can predict correct gradient even when the query information is limited. These results give strong evidence on effectiveness of our proposed method for enhancing query efficiency.

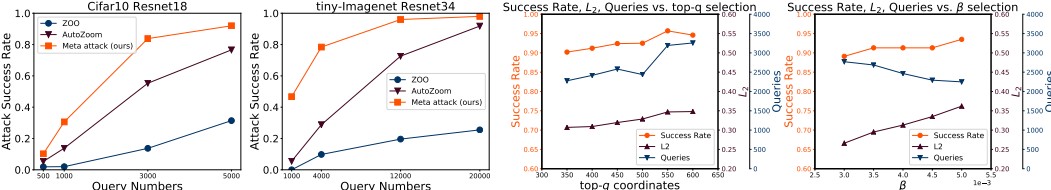

Figure 1: Comparison with limited queries.    Figure 2: Top-$q$ and $\beta$ selection.

**Targeted Attack** Targeted attack aims to generate adversarial noise such that the perturbed sample would be mis-classified into any pre-specified category. It is a more strict setting than the untargeted

Table 1: MNIST, CIFAR10 and tiny-ImageNet untargeted attack comparison: Meta attacker attains comparable success rate and $L_2$ distortion as baselines, and significantly reduces query numbers.

| Dataset / Target model | Method | Success Rate | Avg. $L_2$ | Avg. Queries |
|---|---|---|---|---|
| MNIST / Net4 | Zoo (Chen et al., 2017) | 1.00 | 1.61 | 21,760 |
| | Decision Boundary (Brendel et al., 2018) | 1.00 | 1.85 | 13,630 |
| | Opt-attack (Cheng et al., 2019) | 1.00 | 1.85 | 12,925 |
| | AutoZoom (Tu et al., 2019), | 1.00 | 1.86 | 2,412 |
| | Bandits (Ilyas et al., 2018b) | 0.73 | 1.99 | 3,771 |
| | Meta attack (ours) | 1.00 | 1.77 | **749** |
| CIFAR10 / Resnet18 | Zoo (Chen et al., 2017) | 1.00 | 0.30 | 8,192 |
| | Decision Boundary (Brendel et al., 2018) | 1.00 | 0.30 | 17,010 |
| | Opt-attack (Cheng et al., 2019) | 1.00 | 0.33 | 20,407 |
| | AutoZoom (Tu et al., 2019) | 1.00 | 0.28 | 3,112 |
| | Bandits (Ilyas et al., 2018b) | 0.91 | 0.33 | 4,491 |
| | FW-black (Chen et al., 2018) | 1.00 | 0.43 | 5,021 |
| | Meta transfer (ours) | 0.92 | 0.35 | **1,765** |
| | Meta attack (ours) | 0.94 | 0.34 | **1,583** |
| tiny-ImageNet / VGG19 | Zoo (Chen et al., 2017) | 1.00 | 0.52 | 27,827 |
| | Decision Boundary (Brendel et al., 2018) | 1.00 | 0.52 | 49,942 |
| | Opt-attack (Cheng et al., 2019) | 1.00 | 0.53 | 71,016 |
| | AutoZoom (Tu et al., 2019) | 1.00 | 0.54 | 8,904 |
| | Bandits (Ilyas et al., 2018b) | 0.78 | 0.54 | 9,159 |
| | Meta transfer (ours) | 0.99 | 0.56 | **3,624** |
| | Meta attack (ours) | 0.99 | 0.53 | **3,278** |
| tiny-ImageNet / Resnet34 | Zoo (Chen et al., 2017) | 1.00 | 0.47 | 25,344 |
| | Decision Boundary (Brendel et al., 2018) | 1.00 | 0.48 | 49,982 |
| | AutoZoom (Tu et al., 2019) | 1.00 | 0.45 | 9,770 |
| | Opt-attack (Cheng et al., 2019) | 1.00 | 0.52 | 60,437 |
| | Bandits (Ilyas et al., 2018b) | 0.73 | 0.49 | 9,978 |
| | Meta transfer (ours) | 0.99 | 0.56 | **3,540** |
| | Meta attack (ours) | 0.99 | 0.53 | **3,268** |

one. For fair comparison, we define the target label for each sample —a sample with label $\ell$ gets the target label $(\ell+1) \mod \#classes$. We deploy our meta attacker the same as above. The results on MNIST, CIFAR10 and tiny-ImageNet are shown in Table 2. Similar to results of untargeted attack, we achieve comparable noise and success rate to baselines but with reduced query numbers.

### 4.3 MODEL ANALYSIS

**Meta Training**   We first test the benefits of meta training by comparing performance of a meta-trained attacker with a Gaussian randomly initialized attacker without meta training on the three datasets. Fig. 3 shows their success rate, $L_2$ distortion and query count results for initial success. The meta pre-trained attacker achieves averagely $7\%$ higher success rate with $16\%$ lower $L_2$ distortion and $30\%$ less queries, compared with the randomly initialized one. This justifies the contributions of meta training to enhancing query efficiency and also attack performance.

Guaranteed by fine-tuning, the randomly initialized attacker succeeds over many testing samples. The fine-tuning works like an inner training in meta training. With sufficient fine-tuning iterations, the randomly initialized attacker functions like a well-trained meta attacker. This explains the effectiveness of the randomly initialized meta attacker on many testing samples compromised by more queries. However, it could not predict gradient as accurate as the well-trained meta attacker during earlier iterations. Such inaccuracy leads to larger $L_2$ distortion at the beginning. On the contrary, the meta training process enables the well-trained meta attacker to fast-adapt to current testing samples. These results highlight the significant advantages of our meta model towards to black-box attack. The process of meta training makes it familiar with gradient patterns of various models.

**Generalizability**   Here we show that our meta attacker trained on one dataset can be transferred to other datasets. We conduct this experiment between CIFAR10 and tiny-Imagenet, denoted as *Meta transfer* in Table 1 and 2. We first apply the meta attacker trained on CIFAR10 to attack VGG19, ResNet34 on tiny-Imagenet respectively, which are different from models used for training meta attacker. Note the meta attacker tested on CIFAR10 has no privileged prior and is not familiar with

Table 2: MNIST, CIFAR10 and tiny-ImageNet targeted attack comparison: Meta attack significantly outperforms other black-box methods in query numbers.

| Dataset / Target model | Method | Success Rate | Avg. $L_2$ | Avg. Queries |
|---|---|---|---|---|
| MNIST / Net4 | Zoo (Chen et al., 2017) | 1.00 | 2.63 | 23,552 |
| | Decision Boundary (Brendel et al., 2018) | 0.64 | 2.71 | 19,951 |
| | AutoZoom (Tu et al., 2019) | 0.95 | 2.52 | 6,174 |
| | Opt-attack (Cheng et al., 2019) | 1.00 | 2.33 | 99,661 |
| | Meta attack (ours) | 1.00 | 2.66 | **1,299** |
| CIFAR10 / Resnet18 | Zoo (Chen et al., 2017) | 1.00 | 0.55 | 66,400 |
| | Decision Boundary (Brendel et al., 2018) | 0.58 | 0.53 | 16,250 |
| | AutoZoom (Tu et al., 2019) | 1.00 | 0.51 | 9,082 |
| | Opt-attack (Cheng et al., 2019) | 1.00 | 0.50 | 121,810 |
| | FW-black (Chen et al., 2018) | 0.90 | 0.73 | 6,987 |
| | Meta transfer (ours) | 0.92 | 0.74 | **3,899** |
| | Meta attack (ours) | 0.93 | 0.77 | **3,667** |
| tiny-ImageNet / VGG19 | Zoo (Chen et al., 2017) | 0.74 | 1.26 | 119,648 |
| | AutoZoom (Tu et al., 2019) | 0.87 | 1.45 | 53,778 |
| | Opt-attack (Cheng et al., 2019) | 0.66 | 1.14 | 252,009 |
| | Meta transfer (ours) | 0.55 | 1.37 | **12,275** |
| | Meta attack (ours) | 0.54 | 1.24 | **11,498** |
| tiny-ImageNet / Resnet34 | Zoo (Chen et al., 2017) | 0.60 | 1.03 | 88,966 |
| | AutoZoom (Tu et al., 2019) | 0.95 | 1.15 | 52,174 |
| | Opt-attack (Cheng et al., 2019) | 0.78 | 1.00 | 214,015 |
| | Meta transfer (ours) | 0.69 | 1.40 | **13,435** |
| | Meta attack (ours) | 0.54 | 1.21 | **12,897** |

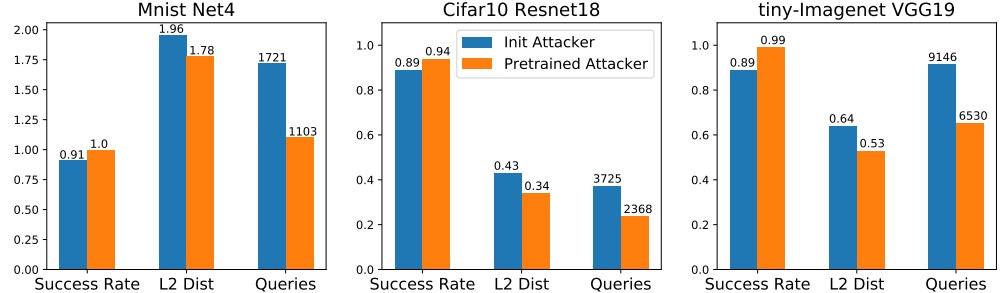

Figure 3: Comparison of randomly initialized and well-trained meta attackers.

neither tiny-Imagenet dataset nor the corresponding classification models. Similarly, we also use the meta attacker trained on tiny-Imagenet to attack the target ResNet18 model on CIFAR10. The results show the good generalizability and robustness of our proposed meta attacker.

**Parameters Selection** We test the selection of top-$q$ and $\beta$ on CIFAR10. We choose $q$ ranging from 350 to 600 and give the results with $\beta$ ranging from 3e-3 to 5e-3, as shown in Fig. 2. When $q$ increases, query number, success rate and $L_2$ will all increase. In order to balance the overall result, we choose $q$ to be 500. When $\beta$ increases, success rate and $L_2$ will increase and query number decreases. In order to balance success rate and query, we choose $\beta$ to be 4e-3 in the experiment.

## 5 CONCLUSION

We propose a meta-based black-box attack method that largely reduces demanded query numbers without compromising in attack success rate and distortion. We train a meta attacker to learn useful prior information about gradient and incorporate it into the optimization process to decrease the number of queries. Specifically, the meta attacker is finetuned to fit the gradient distribution of target model and each update is based on the output of finetuned meta attacker. Extensive experimental results confirm the superior query-efficiency of our method over baselines.

ACKNOWLEDGEMENT

This research is partially supported by Programmatic grant no. A1687b0033 from the Singapore government's Research, Innovation and Enterprise 2020 plan (Advanced Manufacturing and Engineering domain).

Hu Zhang (No. 201706340188) is partially supported by the Chinese Scholarship Council.

Jiashi Feng was partially supported by NUS IDS R-263-000-C67-646, ECRA R-263-000-C87-133, MOE Tier-II R-263-000-D17-112 and AI.SG R-263-000-D97-490.

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

## 6 APPENDIX

### 6.1 MORE EXPERIMENTAL RESULTS

#### 6.1.1 COSINE SIMILARITY BETWEEN ESTIMATED GRADIENTS AND WHITE-BOX GRADIENTS

To demonstrate the ability of estimating gradient of our meta attacker, we have conducted experiments with the Resnet-34 model on the tiny-Imagenet dataset to compare the cosine similarity between the estimated gradients from our proposed meta-attacker and the accurate white-box gradients. We also compare the cosine similarity between ZOO (Chen et al., 2017) estimated gradients and the white-box gradients as reference. The results of cosine similarity and required number of queries are shown in Table 3. We can observe that our meta-attacker does fast-adapt to the target model and generate accurate gradients. Not only it estimates gradients with positive cosine similarity to the true gradients, but it also performs closer to the ZOO estimated results with the same small standard deviation.

Table 3: Cosine similarity between estimated gradients and white-box gradients.

| Task Type | Ours meta-attacker | | ZOO (Chen et al., 2017) | |
|---|---|---|---|---|
| | Similarity | Queries | Similarity | Queries |
| Untargeted | $0.356 \pm 0.074$ | 3,268 | $0.395 \pm 0.079$ | 25,344 |
| Targeted | $0.225 \pm 0.090$ | 12,897 | $0.363 \pm 0.069$ | 88,966 |

#### 6.1.2 VANILLA TRAINING AUTOENCODER AND META-ATTACKER LEARNING FROM ESTIMATED GRADIENTS

We have conducted experiments in section 4.3 to demonstrate the benefits of meta training. However, we only compare the performance of a meta-trained attacker with a Gaussian randomly initialized attacker. We conduct two more experiments on meta training here to further investigate the benefits of meta training. First we compare our meta-trained attacker with a vanilla autoencoder that learns to map images to gradients of one white-box model. The vanilla autoencoder has the same architecture with our meta-attacker, but it is trained in one white-box model. Then, we train a new meta-attacker in four black-box models, i.e., the gradients used for training are estimated via ZOO (Chen et al., 2017). The experiment results are presented in Table 4, 5 .

Table 4: MNIST untargeted attack comparison.

| Method | Success Rate | Avg. $L_2$ | Avg. Queries |
|---|---|---|---|
| Ours(reported) | 1.000 | 1.78 | 1,103 |
| ZOO estimated | 1.000 | 1.77 | 1,130 |
| Vanilla Autoencoder | 1.000 | 1.93 | 1,899 |
| Initialised attacker | 0.912 | 1.96 | 1,721 |

Table 5: MNIST targeted attack comparison.

| Method | Success Rate | Avg. $L_2$ | Avg. Queries |
|---|---|---|---|
| Ours(reported) | 1.000 | 2.66 | 1,971 |
| ZOO estimated | 1.000 | 2.42 | 2,105 |
| Vanilla Autoencoder | 1.000 | 2.80 | 2,905 |
| Initialised attacker | 0.895 | 2.81 | 3,040 |

The two experiments are conducted on 1000 randomly selected images from the MNIST testing set. As for the four approach settings: "Ours (reported)" is the results we report in our paper; "ZOO estimated" is the meta-attacker trained from ZOO estimated gradients; "Vanilla Autoencoder" is the autoencoder maps image to gradient trained in one different MNIST classification model; "Initialised attacker" is the meta-attacker with randomly initialized weights. We can see the "ZOO estimated" model performs closer to our reported meta-attacker. However, the "Vanilla Autoencoder" performs much worse (with larger L2-norm and more queries), which performs similarly to randomly initialized meta-attacker. The two experiments verify the effectiveness of our first meta-training phase.

## 6.2 STRUCTURE OF META ATTACKER

Table 6: **Structure of meta attacker.** Conv: convolutional layer, Convt: de-convolutional layer.

| Meta attacker (MNIST) | Meta attacker (CIFAR10, tiny-ImageNet) |
|---|---|
| Conv(16, 3, 3, 1) + ReLu + bn | Conv(32, 3, 3, 1) + ReLu + bn |
| Conv(32, 4, 4, 2) + ReLu + bn | Conv(64, 4, 4, 2) + ReLu + bn |
| Conv(64, 4, 4, 2) + ReLu + bn | Conv(128, 4, 4, 2) + ReLu + bn |
| Conv(64, 4, 4, 2) + ReLu + bn | Conv(256, 4, 4, 2) + ReLu + bn |
| Convt(64, 4, 4, 2) + ReLu + bn | Convt(256, 4, 4, 2) + ReLu + bn |
| Convt(32, 4, 4, 2) + ReLu + bn | Convt(128, 4, 4, 2) + ReLu + bn |
| Convt(16, 4, 4, 2) + ReLu + bn | Convt(64, 4, 4, 2) + ReLu + bn |
| Convt(8, 3, 3, 1) + ReLu + bn | Convt(32, 3, 3, 1) + ReLu + bn |

## 6.3 STRUCTURE OF TARGET MODEL USED IN MNIST

Table 7: Neural network architecture used on MNIST.

| MNIST Model (Conv: convolutional layer, FC: fully connected layer.) |
|---|
| Conv(128, 3, 3) + Tanh |
| MaxPool(2,2) |
| Conv(64, 3, 3) + Tanh |
| MaxPool(2,2) |
| FC(128) + Relu |
| FC(10) + Softmax |

## 6.4 ACCURACY OF TARGET MODELS ON ORIGINAL DATASETS

Table 8: Accuracy of each target model on each dataset

| Dataset | MNIST | CIFAR10 | tiny-ImageNet | |
|---|---|---|---|---|
| Model | MNIST Model | Resnet18 | VGG19 | Resnet34 |
| Accuracy | 0.9911 | 0.9501 | 0.6481 | 0.6972 |

## 6.5 ADVERSARIAL EXAMPLES GENERATED BY OUR METHOD

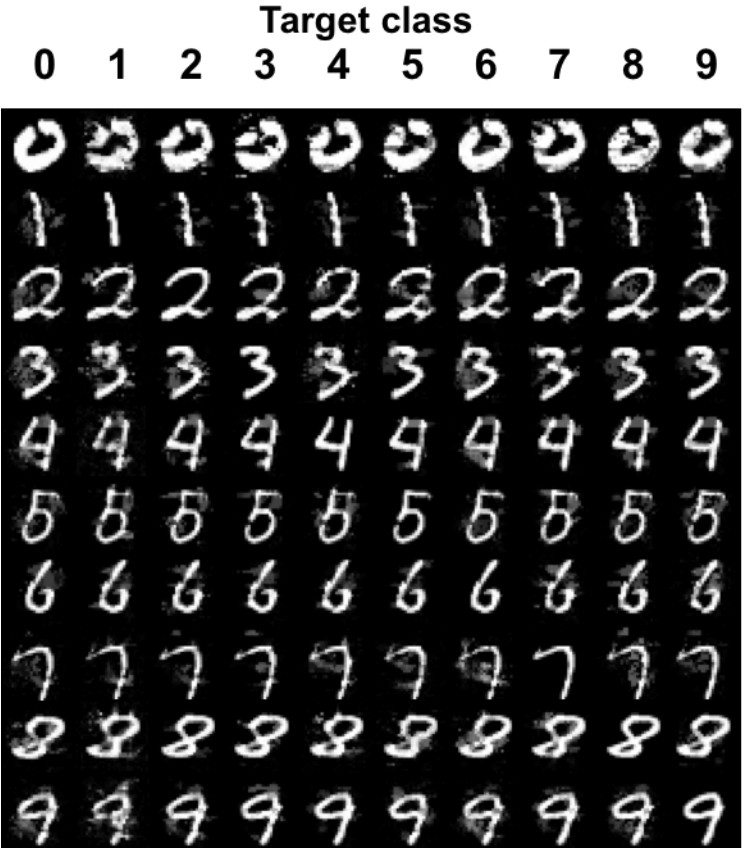

Figure 4: Adversarial examples generated by our method on MNIST. The groundtruth images are shown in the diagonal and the rest are adversarial examples that are misclassified to the targeted class shown on the top.

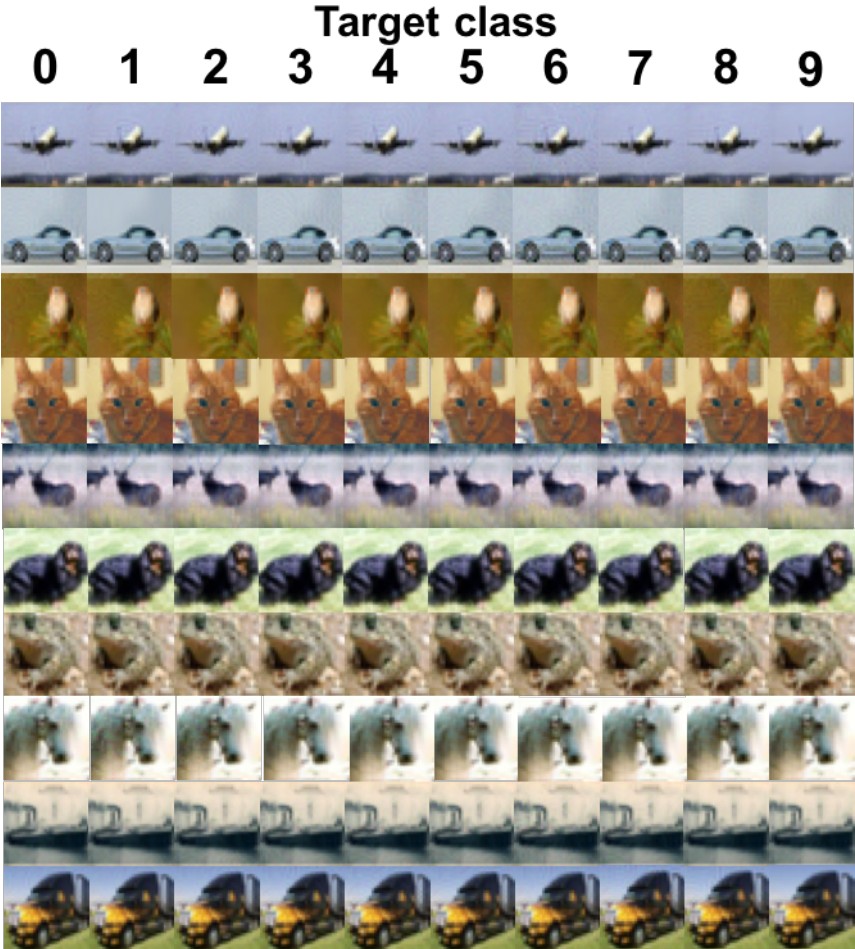

Figure 5: Adversarial examples generated by our method on CIFAR10. The groundtruth images are shown in the diagonal and the rest are adversarial examples that are misclassified to the targeted class shown on the top.

