# OpenReview forum: "Query-efficient Meta Attack to Deep Neural Networks"
_ICLR.cc/2020/Conference — Accept (Poster)_

### Official Review · AnonReviewer3 · 2019-10-22
**Official Blind Review #3**

**Rating:** 6

**Review:**

Summary:

The authors of this paper propose a novel approach for query-efficient black-box adversarial attacks by leveraging meta-learning to approximate the gradient of given clean images.

Paper strength:
1.	The overall idea of this paper is interesting to improve the transferability of adversarial examples through meta-learning.
2.	The paper is well-organized and easy to follow.
3.	Adequate experiment results demonstrate that the meta-attacker is efficient to decrease the number of queries for the adversarial attacks. It is interesting to see the meta attacker trained on certain source domain can be transferred to other domains.

Paper weakness:

1.	The authors miss some important details about training meta attackers like how many images you choose to get the image and its gradient pairs. Are the images from the training set or test set. Is there an overlap between the meta-attacker training set and the test set?
2.	What if you try to learn the adversarial perturbation directly instead of learning the gradient of images?
3.	How about changing the meta learner to vanilla training and learning an autoencoder directly to map the image to its gradient? There could be a problem of how to incorporate the existing classifiers, but I think this could be an important baseline.
4.	The authors try to use the gradient from the ZOO to guide the pre-trained meta attacker to adapt to a new classifier in Algorithm 2 (line 3). What if you take multi-step ZOO to give a much stronger prior to finetuning?
5.	Take a look at the confusion matrix in [1] which studies the transferability of white-box attack and black-box attack. I think it could be more interesting if the authors change the ground truth from gradient to estimated gradient from ZOO for learning meta-attacker.


Li, Yandong, et al. "NATTACK: Learning the Distributions of Adversarial Examples for an Improved Black-Box Attack on Deep Neural Networks." arXiv preprint arXiv:1905.00441 (2019).


**Experience Assessment:**

I have published one or two papers in this area.

**Review Assessment: Checking Correctness Of Derivations And Theory:**

I assessed the sensibility of the derivations and theory.

**Review Assessment: Checking Correctness Of Experiments:**

I assessed the sensibility of the experiments.

**Review Assessment: Thoroughness In Paper Reading:**

I read the paper at least twice and used my best judgement in assessing the paper.

---

> ### Author Response · Authors · 2019-11-13
> **Response (AnonReviewer3) -Part 1**
>
> Thank you for your thoughtful review. We have conducted three additional experiments to answer your questions; our response is as below.
>
> Question 1: The authors miss some important details about training meta attackers like how many images you choose to get the image and its gradient pairs. Are the images from the training set or test set. Is there an overlap between the meta-attacker training set and the test set?
>
> Answer: We are sorry that we did not indicate the details of meta training. We have updated the details of meta-training in experiment section of our paper. For the MNIST, CIFAR10 and tiny-Imagenet datasets, we only use 10k randomly selected images from the training set, the proportion of the selected images and its gradient pair to the whole training set are 16%, 20%, and 10% respectively. During the attack phase, we test the images from the testing set. Therefore, there is no overlap between the meta-attacker training set and the testing set.
>
> =======================
>
> Question 2: What if you try to learn the adversarial perturbation directly instead of learning the gradient of images?
>
> Answer:  We attempted to train the meta-attacker to generate the  (1) adversarial perturbation  (2) gradients of input images at the first phase of our work. There are mainly two reasons that we gave up learning adversarial perturbations directly. First, adversarial perturbations have many solutions to one clean image. Different attack methods and different $l_p$ norm constraints lead to enormous possible adversarial perturbations, which makes adversarial perturbations hard to learn. Second, most of the attack methods focus on how to utilize gradients. The meta-attacker could be combined with these optimizing method if it learns to generate gradients.
>
> =======================
>
> Question 3, 5:
> How about changing the meta learner to vanilla training and learning an autoencoder directly to map the image to its gradient? There could be a problem of how to incorporate the existing classifiers, but I think this could be an important baseline.
> Take a look at the confusion matrix in [1] which studies the transferability of white-box attack and black-box attack. I think it could be more interesting if the authors change the ground truth from gradient to estimated gradient from ZOO for learning meta-attacker.
>
> Answer: Thanks a lot for the constructive review. We have conducted two experiments in MNIST datasets responding to question 3 and question 5.  The first experiment is train a vanilla  autoencoder to generate gradients.  The training MNIST classification model is different from the target classification model in the testing phase.  The second experiment is meta-training a meta-attacker from gradients estimated from ZOO. That is, the training gradients are not generated by the white-box method, but estimated by the black-box ZOO method. The experiment results are presented as follows.
>
> Untargeted                 |Attack Success Rate | L2 Norm | Number of Queries
> ———————————————————————————————————
> Ours (reported)         |     1.000                      |   1.78       |  1,103
> ———————————————————————————————————
> ZOO estimated          |      1.000                     |   1.77       |  1,130
> ———————————————————————————————————
> Vanilla Autoencoder |      1.000                     |   1.93       |  1,899
> ———————————————————————————————————
> Initialised attacker    |      0.912                     |   1.96       |  1,721
> ———————————————————————————————————
>
> targeted                     |Attack Success Rate| L2 Norm | Number of Queries
> ———————————————————————————————————
> Ours (reported)         |    1.000                     |    2.66     |    1,971
> ———————————————————————————————————
> ZOO estimated          |    1.000                      |   2.42     |    2,105
> ———————————————————————————————————
> Vanilla Autoencoder |     1.000                     |   2.80     |    2,905
> ———————————————————————————————————
> Initialised attacker    |     0.895                     |    2.81    |    3,040
> ———————————————————————————————————
>
> The two experiments are conducted on 1000 randomly selected images from the MNIST testing set. As for the four approach settings: "Ours (reported)" is the results we report in our paper; "ZOO estimated" is the meta-attacker trained from ZOO estimated gradients; "Vanilla Autoencoder" is the autoencoder maps image to gradient trained in one different MNIST classification model; "Initialised attacker" is the meta-attacker with randomly initialized weights. We can see the "ZOO estimated" model performs closer to our reported meta-attacker. However, the "Vanilla Autoencoder" performs much worse (with larger L2-norm and more queries), which performs similarly to randomly initialized meta-attacker. The two experiments verify the effectiveness of our first meta-training phase. We have updated the two experiments results in appendix 6.1.2 of our paper.
>
> =======================

---

> ### Author Response · Authors · 2019-11-13
> **Response (AnonReviewer3) -Part 2**
>
> Cont.
>
> Question 4: The authors try to use the gradient from the ZOO to guide the pre-trained meta attacker to adapt to a new classifier in Algorithm 2 (line 3). What if you take multi-step ZOO to give a much stronger prior to fine-tuning?
>
> Answer: One significant drawback of stronger prior is that the number of queries will increase substantially.  Moreover, a stronger prior may decrease the performance of the meta-attacker. The reason is that our meta-attacker is trained with clean images to achieve fast-adaptive. During the attacking phase, the meta-attacker cannot generate accurate gradients as the magnitude of adversarial perturbation increases. That is why we add the finetuning step for every $m$ iteration in Algorithm 2. We want a not so strong but estimate every $m$ iteration to make the meta-attacker fast adapted to current unseen adversarial examples with less required queries.

---

### Official Review · AnonReviewer2 · 2019-10-23
**Official Blind Review #2**

**Rating:** 8

**Review:**

The authors proposed a meta-learning based black-box adversarial attack method to reduce the number of queries. Their proposed meta attacker is trained via a meta-learning approach, and it could utilize historical query information and fast adapted to different target models. The authors conduct experiments in three datasets, and the results show that their proposed meta attacker is more efficient than other black-box attack methods.

Strong points of this paper:
1) novelty. The paper combines meta-learning and black-box attack and develops a deep convolutional model (meta attacker) to predict the gradients of another DNN model. Most of the other query-efficient attack methods focus on utilizing estimated gradients, while this paper focuses on predicting accurate gradients. That makes this paper novel.
2) The results are good. The proposed method could attain comparable l2 norm and attack success rate with much fewer queries. And the analysis in experiments is interesting. The generalizability experiment of the meta-attacker(meta transfer) shows that gradients in different models and different datasets have some similar patterns.

Weak points of this paper:
1) The authors could investigate more in the theoretical part. For example, the authors could gives the theoretical support of why a convolutional network could stimulate other networks' gradient.
2) The deploy of the meta-attacker is not as easy as other black-box attack methods. It would not be practical if we need to meta training the attacker before we use it.

Questions:
1) Is it hard to meta training the meta-attacker? How long will it take for training?
2) Is it necessary to finetune the meta attacker in algorithm 2 every m iteration? How to determine the "m"?
3) Could the proposed meta-attacker embedded with other optimized black-box attack methods?

Overall, this paper is well-structured, novel, and ideas are well motivated. The query-efficient black-box attack problem is practical and meaningful. I would encourage the authors to release their codes. Last, I would recommend acceptance for this paper.



**Experience Assessment:**

I have published one or two papers in this area.

**Review Assessment: Checking Correctness Of Derivations And Theory:**

I carefully checked the derivations and theory.

**Review Assessment: Checking Correctness Of Experiments:**

I carefully checked the experiments.

**Review Assessment: Thoroughness In Paper Reading:**

I read the paper thoroughly.

---

> ### Author Response · Authors · 2019-11-13
> **Response (AnonReviewer2)**
>
> We want to thank the reviewer for the positive assessment. We will release our codes and pre-trained meta-attacker. Our responses to the weak points and questions are as below.
>
> Weak point: The deploy of the meta-attacker is not as easy as other black-box attack methods. It would not be practical if we need to meta training the attacker before we use it.
>
> Answer: We agree with that the meta-attacker is not as convenient as other numerical estimation and optimization of gradients black-box attack methods. We realized that inconvenience and therefore proposed the meta-transfer method in our paper. The meta-transfer model could be applied to datasets that share similarities (cifar10 and tiny-Imagenet for example) with minimal performance loss and offers appealing convenience. We will release the pre-trained meta-transfer models as well as our codes.
>
> =======================
>
>
> Question 1: Is it hard to meta training the meta-attacker? How long will it take for training?
>
> Answer: The meta training of the meta is easy and fast. It takes about half an hour of training time in MNIST and CIFAR 10 datasets, and about two hours in tiny-Imagenet datasets. We conducted our experiments on one Nvidia Titan X (Pascal) GPU.
>
> =======================
>
>
> Question 2: Is it necessary to finetune the meta attacker in algorithm 2 every m iteration? How to determine the "m"?
>
> Answer: Yes, it is necessary. During the iterative steps to construct the adversarial examples, the current perturbed examples are very different from the clean images, which makes the meta-attacker trained with clean examples hard to predict the gradients. Fine-tuning the meta attacker makes the attacker fast adapted to current perturbed examples.
> We conducted additional untargeted attack experiments on the Resnet-34 tiny-Imagenet model to demonstrate the effects of different "m." The results are presented below.
>
> m          |   Attack Success Rate        | L2 Norm    | Number of Queries
> —————————————————————————————
> 2           |          0.99                             |       0.52	     |       9,505
> —————————————————————————————
> 3           |	         0.98                             |       0.54       |       6,826
> —————————————————————————————
> 4           |          1.00                             |       0.66       |       6,307
> ————————————————————————————---
> 5           |          1.00                             |       0.73       |       5,504
> ————————————————————————————---
>
> As we can see from the table, if we finetune more frequently, the l2 norm will decrease but with many more queries; if we finetune less frequently, the l2 norm will increase but with fewer queries. To balance the l2 norm and number of queries, we choose m = 3 for the tiny-Imagenet datasets.
>
> =======================
>
> Question 3: Could the proposed meta-attacker embedded with other optimized black-box attack methods?
>
> Answer: Yes, and that is a critical advantage of our proposed meta-attacker. Most of the black-box attack methods are either numerical estimation or gradient optimization methods. Our method is to meta-train a model to predict gradients directly. Therefore, our method could be combined with numerical estimation methods in finetuning and with gradient optimization methods in applying gradients.  For instance, we used the Adam-like method to apply predicted gradients.

---

> > ### Comment · AnonReviewer2 · 2019-11-15
> > **raise rating**
> >
> > The authors have successfully addressed the majority of my concerns. The additional experiment of comparing m answers my question well. I would like to raise my rating.

---

### Official Review · AnonReviewer4 · 2019-10-28
**Official Blind Review #4**

**Rating:** 6

**Review:**

This paper proposed a meta attack approach that is capable of attacking a targeted model with fewer queries. It effectively utilizes meta learning approaches to abstract generalizable prior from previous observed attack patterns. Experiments on several benchmark datasets demonstrates it performance in reducing model queries.

- The proposed meta learning approach seems to learn a universal gradient direction of an image x for all networks, which seems to be an ambitious goal. The authors did not provide any intuition or demonstrative explanations towards this. I hope the author could provide some more evidence showing that why this is achievable and is this indeed achieved by the meta learner (e.g., comparing the cosine similarity between the true gradient and the meta learner generated gradient?)

- In Algorithm 2, the authors seem to use coordinate-wise zeroth-order gradient estimation as in ZOO. I wonder have the authors considered using NES type Gaussian noise to estimate the gradient? As it has been shown to be more efficient than coordinate-wise zeroth-order gradient estimation in (Ilyas et al. 2018).

- I notice that the authors only consider L2 norm attack case and did not include more popular L-infinity norm case. Is there any reason why it can not be applied to L-infinity norm? I didn’t find anywhere in the algorithm that would restrict the choice of norm type. The author might also need to compare with the following recent papers regarding black-box attacks.

Yan, Ziang, Yiwen Guo, and Changshui Zhang. "Subspace Attack: Exploiting Promising Subspaces for Query-Efficient Black-box Attacks." arXiv preprint arXiv:1906.04392 (2019).
Moon, Seungyong, Gaon An, and Hyun Oh Song. "Parsimonious Black-Box Adversarial Attacks via Efficient Combinatorial Optimization." ICML 2019.
Chen, Jinghui, Jinfeng Yi, and Quanquan Gu. "A Frank-Wolfe Framework for Efficient and Effective Adversarial Attacks." arXiv preprint arXiv:1811.10828 (2018).

- In experiments part, only 100 images from each dataset may not be representative enough. I would suggest the authors to test more samples. Also please consider only the images that can be correctly classified without perturbation, as there is no need to attack those already misclassified images.


**Experience Assessment:**

I have published one or two papers in this area.

**Review Assessment: Checking Correctness Of Derivations And Theory:**

N/A

**Review Assessment: Checking Correctness Of Experiments:**

I assessed the sensibility of the experiments.

**Review Assessment: Thoroughness In Paper Reading:**

I read the paper at least twice and used my best judgement in assessing the paper.

---

> ### Author Response · Authors · 2019-11-13
> **Response (AnonReviewer4) - Part 1**
>
> Thanks a lot for the review and the helpful advice. We answer the questions as follows:
>
> Question 1: The proposed meta learning approach seems to learn a universal gradient direction of an image x for all networks, which seems to be an ambitious goal. The authors did not provide any intuition or demonstrative explanations towards this. I hope the author could provide some more evidence showing that why this is achievable and is this indeed achieved by the meta learner (e.g., comparing the cosine similarity between the true gradient and the meta learner generated gradient?)
>
> Answer: The most important feature of our proposed meta-attacker is its fast-adaptivity, i.e. learning to predict the full target model gradients based on a few numerical estimates (i.e. from ZOO). It is not a universal gradient estimator. However, as long as two independent datasets share similarities in feature space our meta-attacker shows good transferability and can fast-adapted to the datasets.
>
> To demonstrate the ability of estimating gradient of our meta attacker, we have conducted experiments with the Resnet-34 model on the tiny-Imagenet dataset to compare the cosine similarity between the estimated gradients from our proposed meta-attacker and the accurate white-box gradients. The results are as below:
>
> Task type      |meta-attacker gradients  |Queries|  ZOO[1] gradients |Queries
> —————————————————————————————————————
> Untargeted  |  $0.356\pm0.074$             |  6,530  | $0.395\pm0.079$  | 25,344
> —————————————————————————————————————
> Targeted      |  $0.225\pm0.090$             | 19,516 | $0.363\pm0.069$  | 88,966
> —————————————————————————————————————
>
> The second and fourth columns in the table are the mean and std of the cosine similarity with the white-box gradients on 1000 testing samples. From the results, we can see that our meta-attacker does fast-adapt to the target model and generate accurate gradients. Not only it estimates gradients with positive cosine similarity to the true gradients, but it also performs closer to the ZOO estimated results with the same small standard deviation. We have updated the experiment results in appendix 6.1.1 of our paper.
>
>
> =======================
>
> Question 2, 3:
> In Algorithm 2, the authors seem to use coordinate-wise zeroth-order gradient estimation as in ZOO[1]. I wonder have the authors considered using NES type Gaussian noise to estimate the gradient? As it has been shown to be more efficient than coordinate-wise zeroth-order gradient estimation in (Ilyas et al. 2018).
> I notice that the authors only consider L2 norm attack case and did not include more popular L-infinity norm case. Is there any reason why it can not be applied to L-infinity norm? I didn’t find anywhere in the algorithm that would restrict the choice of norm type. The author might also need to compare with the following recent papers regarding black-box attacks.
>
> Answer: Yes, our proposed method can using NES as the reference gradient estimation for fine-tuning and can also perform well in $L_\infty$ norm constraints.
> Since our proposed meta-attacker derived from ZOO [1], for fair comparison with other methods on query efficiency, we take the same $L_2$ norm constraints as ZOO[1] for ensuring evaluation metric to be consistent. Therefore, we also compare our results with other baselines using  $L_2$ norm and attack success rate. We attempted to use NES as the reference gradients estimation for fine-tuning our meta-attacker, but it leads to much higher $L_2$ norm of the perturbation than  ZOO [1]. The following results in Resnet34 tiny-Imagenet model are from taking NES for untargeted attack over 1000 randomly selected images.
>
> Type           |Attack Success Rate   | L2 Norm | Number of Queries
> ———————————————————————————————
> Ours(ZOO)|    0.98                          |   0.49        |    6,866
> ———————————————————————————————
> Ours(NES) |    0.82                          |   1.93        |    5,586
> ———————————————————————————————
> ZOO           |    1.00                          |   0.47        |   25,334
> ———————————————————————————————
>
> From the results, although NES requires fewer queries, it will lead to lower attack success rate and  much larger $L_2$ norm.

---

> > ### Comment · AnonReviewer4 · 2019-11-14
> > **Official Blind Review #4**
> >
> > I have read the response and I think it well-addressed my concerns. I will increase my score.

---

> ### Author Response · Authors · 2019-11-13
> **Response (AnonReviewer4) - Part 2**
>
> Cont.
> Answer to Question 2,3:
> One advantage of our proposed meta-attacker is that it can be easily combined with other black-box gradient estimation methods or gradient optimization methods. In the next experiment, we compare with the suggested two baselines. We take the NES as the gradient estimation reference under $L_\infty$ norm constraint.
>
> Due to the time limit, we compare our proposed method with the two baselines [2,4]. We compare our proposed method on the same public CIFAR10 model WRN to [2]. We tested the released codes of [4] for comparison with ours under the same experiment. The following results show that our proposed meta-attacker outperforms [2,4]. We have updated [4] results on CIFAR10 in our paper. We will add the mentioned three baselines [2,3,4] in the next version of our paper.
>
> Untargeted CIFAR10 , WRN model, $L_\infty$ norm with $\epsilon = 8/255$
> Type            |Attack Success Rate |   Number of Queries
> —————————————————————————————
> Ours(NES)  |    0.991                      |        361
> —————————————————————————————
> [2]                |    0.997                     |        389
> —————————————————————————————
>
> Untargeted CIFAR10
> Type     |Attack Success Rate |   L2 Norm |Number of Queries
> —————————————————————————————
> Ours       |    0.92                      |    0.35      |  2,438
> —————————————————————————————
> [4]           |    1.00                      |    0.43      |  5,021
> —————————————————————————————
>
>
>
> =======================
>
> Question 4: In experiments part, only 100 images from each dataset may not be representative enough. I would suggest the authors to test more samples. Also please consider only the images that can be correctly classified without perturbation, as there is no need to attack those already misclassified images.
>
> Answer: All the reported results in our paper are conducted only on the clean images that are correctly classified by the target model. Following the suggestion, we tested our proposed method in 1000 randomly selected images from the testing set, and the results are shown in the following table.
> tiny-Imagenet
> Untargeted attack                |Attack Success Rate | L2 Norm | Number of Queries
> ——————————————————————————————————
> Resnet34 (reported)            |    0.98                         |    0.49     |    6,866
> ——————————————————————————————————
> Resnet34 (1000 samples)   |    0.993                       |   0.53      |    6,498
> ——————————————————————————————————
> VGG19 (reported)                |    0.98                         |    0.54     |    6,826
> ——————————————————————————————————
> VGG19 (1000 samples)       |    0.994                       |   0.53      |    6,530
> ——————————————————————————————————
>
> From the above results, our proposed meta-attacker is robust and performs consistently well for large-scale test samples. We have updated the new untargeted results on 1000 samples in our paper, we will update the rest targeted results on 1000 samples once the experiments are finished.
>
> Reference:
> [1] Pin-Yu Chen, Huan Zhang, Yash Sharma, Jinfeng Yi, and Cho-Jui Hsieh. Zoo: Zeroth order optimization based black-box attacks to deep neural networks without training substitute models. In Proceedings of the 10th ACM Workshop on Artificial Intelligence and Security, pp. 15–26. ACM, 2017.
> [2] Yan, Ziang, Yiwen Guo, and Changshui Zhang. "Subspace Attack: Exploiting Promising Subspaces for Query-Efficient Black-box Attacks." arXiv preprint arXiv:1906.04392 (2019).
> [3] Moon, Seungyong, Gaon An, and Hyun Oh Song. "Parsimonious Black-Box Adversarial Attacks via Efficient Combinatorial Optimization." ICML 2019.
> [4] Chen, Jinghui, Jinfeng Yi, and Quanquan Gu. "A Frank-Wolfe Framework for Efficient and Effective Adversarial Attacks." arXiv preprint arXiv:1811.10828 (2018).

---

### Decision · Program_Chairs · 2019-12-19

**Decision:**

Accept (Poster)

**Comment:**

This paper proposes a meta attack approach based on meta learning approaches to learn generalizable prior from the previously observed attack patterns. The proposed approach is able to attack targeted models with much fewer queries. After author response, all reviewers are very positive about the paper. Thus I recommend accept.